# Simulating Human Gaze with Neural Visual Attention

**Leo Schwinn**[1,2]    **Doina Precup**[2,3,4]    **Bjoern M. Eskofier**[1]    **Dario Zanca**[1]

[1]Friedrich-Alexander-Universität Erlangen-Nürnberg    [2]Mila    [3]McGill University    [4]DeepMind

Corresponding author: `dario.zanca@fau.de`

## Abstract

Existing models of human visual attention are generally unable to incorporate direct task guidance and therefore cannot model an intent or goal when exploring a scene. To integrate guidance of any downstream visual task into attention modeling, we propose the Neural Visual Attention (NeVA) algorithm. To this end, we impose to neural networks the biological constraint of foveated vision and train an attention mechanism to generate visual explorations that maximize the performance with respect to the downstream task. We observe that biologically constrained neural networks generate human-like scanpaths without being trained for this objective. Extensive experiments on three common benchmark datasets show that our method outperforms state-of-the-art unsupervised human attention models in generating human-like scanpaths.

Full paper available at TMLR:
`https://openreview.net/forum?id=7iSYW1FRWA`.

## 1 Introduction

Computational modeling of human visual attention lies at the intersection of many disciplines, such as neuroscience, cognitive psychology, and computer vision. Despite eye-tracking technologies becoming increasingly cost-efficient, collecting human expert gaze data in domain-specific applications, e.g., medical or higher education fields, remains expensive. Therefore, models simulating plausible task-specific scanpaths are highly required both for understanding the biological mechanism [14, 21], as well as in applications [19, 10, 18, 7].

Most eye-tracking datasets and related methods are designed for saliency prediction [2, 23]. However, saliency maps are static and cannot describe the temporal aspect of visual exploration. On the other end, scanpath models have been proposed that can generate fixation sequences. The circuitry of winner-take-all [14] can be combined with any saliency estimation method, such as Itti's saliency map [11], to generate scanpaths in an unsupervised manner. Boccignone et al. [3] generate gaze trajectories based on non-local transition probabilities defined in a saliency field. Recently, [22] described attention by mean of gravitational laws of motion, where visual features are considered as masses attracting the focus of attention. However, existing approaches tacitly assume perfect vision by processing the input in its full resolution all at once, and do not provide a flexible way to incorporate task guidance to the generated scanpaths.

We propose a Neural Visual Attention (NeVA) algorithm to generate purely task-driven gaze trajectories. The algorithm consists of three major components. First, a differentiable foveation layer that, given a fixation position and an input image, simulates biologically plausible foveated vision. Second, a task model that gets passed a foveated image by the foveation layer and produces a loss signal with respect to its downstream task (e.g., classifying the original class or reconstructing the original image). Lastly, an attention mechanism determining the next fixation position in order to minimize

Gaze Meets ML Workshop at the 36th Conference on Neural Information Processing Systems (NeurIPS 2022).

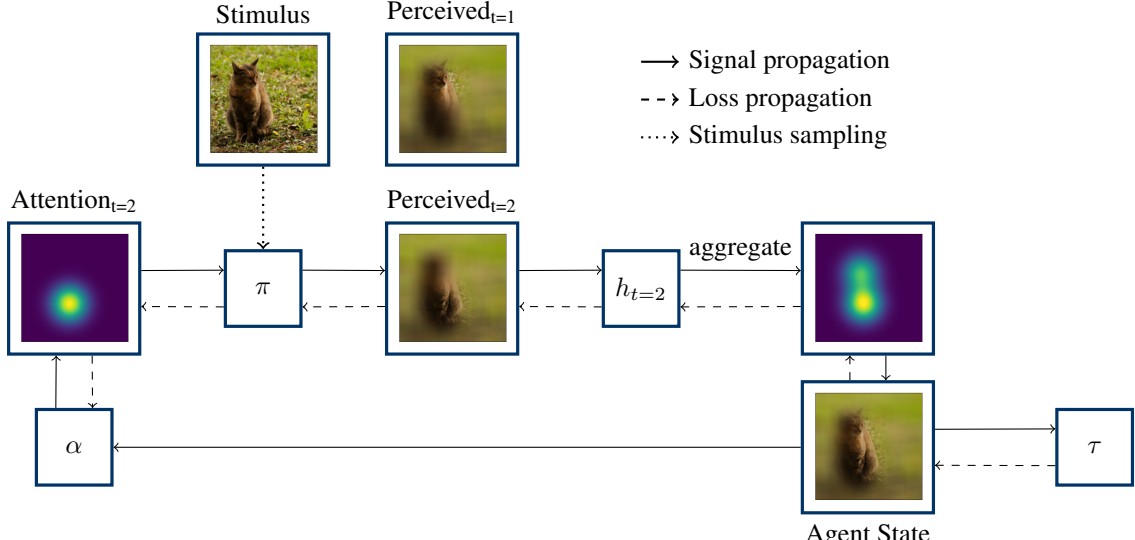

Figure 1: The Neural Visual Attention (NeVA) algorithm.

the loss with respect to the task model (see Figure 1 for an illustration). The resulting architecture is fully differentiable. We use neural networks to model both the attention and the task model and backpropagation to train the attention mechanism. The task model is only used to generate the loss during training and can be discarded at inference time. The flexibility of the framework allows us to analyze the contribution of different tasks and hence generate task-specific gaze trajectories. To validate the proposed approach, we compare it to three well-established unsupervised models of visual attention and three baseline approaches. The plausibility of the models is measured as the similarity of the generated scanpath with human eye-tracking data. Results demonstrate the superiority of the proposed approach.

## 2 Neural Visual Attention (NeVA)

We propose an algorithm for Neural Visual Attention (NeVA) which enables to generate scanpaths of visual attention under the guidance of any differentiable task model. As previously mentioned, this is made by three main components: a task model, a foveation layer, and an attention mechanism. In what follows, we formally define each component of the model. The approach is illustrated in Figure 1.

**Task model.** Let $S = \{S_1, ..., S_N\}$ be a collection of input stimuli, and $Y = \{y_1, ..., y_N\}$ be a set of corresponding targets (i.e., they correspond to class labels in the case of a classification task). A task model $\tau$ is trained to map input stimuli to their corresponding targets, such that

$$\tau(S_i) = y_i, \forall i \in \{1, ..., N\}.$$

**Foveation layer.** A fully differentiable foveation layer is defined to simulate human vision, constrained by its structure to fine vision in corresponding to the fovea, and coarse vision in the periphery. Given a stimulus $S$, the foveation mechanism computes the perceived stimulus $\pi(S, \xi_t)$ as a foveated rendering of $S$ centered at the current focus of attention $\xi_t$. In particular, let $\tilde{S}$ be a coarse version of $S$, obtained by applying a convolution to the original stimulus $S$ to suppress any high frequency components. Then, the foveated stimulus is obtained as a linear combination of the original stimulus and its coarse version,

$$\pi(S, \xi_t) = G_{\sigma_\xi}(t) \cdot S + (1 - G_{\sigma_\xi}(t)) \cdot \tilde{S},$$

where $G_{\sigma_\xi}(t)$ is a Gaussian blob, with mean $\xi_t$ and standard deviation $\sigma_\xi$, centered at $\xi_t$.

To incorporate past information, we define the internal *agent's state* $h_t \sim h(S, \xi_t)$ to express the cumulative perceived information, or *memory* of the system, such that

$$h(S, \xi_t) = G_\Sigma(t) \cdot S + (J_d - G_\Sigma(t)) \cdot \tilde{S},$$

with

$$G_\Sigma(t) = \left[\sum_{i=0}^\infty \gamma^i G_{\sigma_\xi}(t-i)\right]^{0:1},$$

where $[...]^{0:1}$ is an element-wise clipping operator with minimum 0 and maximum 1, and $J_d$ is a $d \times d$ dimensional unit matrix (matrix only filled with ones). The parameter $\gamma \in [0,1]$ can be regarded as a forgetting coefficient.

**Attention mechanism.** A mechanism of attention is trained to generate the next location of focus of attention, based on the current internal representation, i.e.,

$$\alpha\left(h\left(S,\xi_t\right)\right) \mapsto \xi_{t+1}.$$

Since we aim at developing a purely task-driven mechanism of attention, parameters of the attention model are optimized to minimize the loss function

$$\mathcal{L}(\tau\left(h\left(S,\xi_t\right)\right), y).$$

The optimal attention mechanism would unblur the regions that lead to the best performance with respect to the underlying vision task.

**Inference with NeVA.** The task model is only used to generate the loss during training, and can be discarded at inference time. The attention mechanism can be used iteratively, to infer the next position of the focus of attention, based on the updated agent's state.

## 3 Experiments

The NeVA algorithm is used to generate scanpath of visual attention. The plausibility of these scanpaths is measured through similarity metrics with human eye-tracking data. We stress out that eye-tracking data is not used during the training of NeVA, but only for evaluation. The performance of NeVA is compared with that of three unsupervised human attention models and three baselines.

**Datasets.** A collection of three well-established eye-tracking dataset is used in our study: MIT1003 [12] (1003 images, 15 subjects, free viewing condition), TORONTO [6] (120 images, 20 subjects, free viewing condition), and KOOTSTRA [15] (99 images, 31 subjects, free viewing condition).

**Metrics.** Similarity with respect of human eye-tracking data is measured using two metrics. The string-edit distance (SED) [13] has been adapted in the domain of visual attention analysis to compare visual scanpaths [5, 9], after being properly converted to strings. Additionally, we define string-based time-delay embeddings (SBTDE) that computes the time delay embedding [1] in the string domain. This metric better take into account the stochastic component of the attention process [17], and making the results more robust with respect to changes in both stimulus resolution and scanpath length. Metrics are presented in two versions: *Mean* metrics are computed by simply averaging the scores with respect to all available subjects for a given image, while *ScanPath Plausibility (SPP)* metrics only considers the scanpath of minimal distance [8].

**Competitors and baselines.** For our experimental comparison, we restrict our comparison to unsupervised approaches, i.e., existing approaches that similarly to us do not use any eye-tracking data to train their models. In particular, we include Constrained Levy Exploration (CLE) [3], Gravitational Eye Movements Laws (G-EYMOL) [22], Winner-take-all (WTA) [14]. CLE and WTA are based on saliency maps by [11]. For all competitors, we use the python implementation provided by the authors in the original paper. Additionally, we define three baselines which help in better positioning the results. For the *Random* baseline, scanpaths are generated as sequences of random fixation points. For the *Center* baseline, subsequent fixations are sampled according to a center blob, as described in [12]. We finally regard the *Human* baseline as the gold standard, where we measure as each human is a good predictor for the remaining population on certain image.

**NeVA versions.** We test two different NeVA configurations. NeVA$_C$ is based on a classification task. A ResNet that was trained on the CIFAR10 dataset [16] from the RobustBench library. NeVA$_R$ is based on a reconstruction-task model. In this case, we train a denoising autoencoder on the CIFAR10 dataset using the implementation proposed in [24]. For both cases, attention models is based on wide ResNets [20].

Table 1: Similarity of scanpaths generated by the proposed NeVA method and other competitors to those of humans for several metrics. A lower score in each metric corresponds to a higher similarity to human scanpaths. The best results are shown in **bold** and the second best results are underlined. The human column denotes the intra-scanpath distance between humans.

| Datasets | NeVA$_C$ | NeVA$_R$ | G-Eymol | CLE | WTA | Center | Random | Human |
|---|---|---|---|---|---|---|---|---|
| **MIT1003** | | | | | | | | |
| Mean SED | **4.3** | 4.49 | 4.48 | 4.60 | 4.90 | 4.99 | 5.09 | 3.74 |
| SPP SED | **3.15** | 3.49 | 3.39 | 3.41 | 4.15 | 4.26 | 4.44 | 1.65 |
| Mean SBTDE | **0.62** | 0.67 | 0.68 | 0.72 | 0.76 | 0.78 | 0.81 | 0.57 |
| SPP SBTDE | **0.51** | 0.57 | 0.59 | 0.60 | 0.67 | 0.71 | 0.74 | 0.23 |
| **TORONTO** | | | | | | | | |
| Mean SED | **4.22** | 4.42 | 4.52 | 4.56 | 4.74 | 5.05 | 5.19 | 3.72 |
| SPP SED | **3.35** | 3.71 | 3.79 | 3.74 | 4.17 | 4.51 | 4.71 | 2.28 |
| Mean SBTDE | **0.58** | 0.64 | 0.69 | 0.72 | 0.73 | 0.82 | 0.85 | 0.64 |
| SPP SBTDE | **0.58** | 0.64 | 0.68 | 0.72 | 0.73 | 0.82 | 0.84 | 0.30 |
| **KOOTSTRA** | | | | | | | | |
| Mean SED | **4.66** | 4.75 | 4.67 | 4.89 | 4.99 | 4.99 | 5.08 | 4.26 |
| SPP SED | **2.98** | 3.26 | 3.12 | 3.12 | 3.66 | 3.75 | 3.88 | 1.16 |
| Mean SBTDE | **0.71** | 0.73 | 0.74 | 0.76 | 0.77 | 0.78 | 0.80 | 0.68 |
| SPP SBTDE | 0.43 | 0.48 | 0.51 | **0.41** | 0.53 | 0.58 | 0.60 | 0.20 |

**Results.** For each model and baseline, we generated scanpaths of length 10, i.e., a sequence of 10 fixations. Table 1 summarizes the results for all metrics and datasets. NeVA$_C$ is the best performing method in $11/12$ metrics and the second-best in the remaining 1. NeVA$_R$ is the second best method in $7/12$ metrics. G-Eymol is the second-best method in 4 metrics, while CLE is the best method in 1 metric and the second-best method in 1 metric (same score as G-Eymol). Unlike competitors, NeVA-based approaches are merely driven by their top-down signal (purely task-driven), and this allows us to directly compare the tasks involved. In fact, we notice that classification tasks better explain human behavior over all three datasets. Moreover, for scanpath of length 10, the NeVA top-down approach produces better results than bottom-up counterparts, supporting the hypothesis that task guidance already emerges in the first phases (within 3 seconds) of visual explorations. The Center baseline did not perform much better than Random. This is a surprising result if we consider that a center blob can predict saliency very well [12, 4]. Human baseline, instead, outperforms all approaches by a large margin. This is particularly true in the case of the SPP versions of the metrics, where only the closest human scanpath is considered for the metric calculation. This result suggests that there is still large margin of improvement, especially in the development of personalized models of attention, which can take into account the large intra-population differences.

## 4   Discussion

In an empirical study, we demonstrate that neural network models pre-trained for image classification or reconstruction can provide effective guidance for generating biologically-plausible visual scanpaths. Such scanpaths resemble human scanpaths, although they were neither explicitly trained for this purpose nor did we use eye movement data during training.

A future study should examine whether the similarity between human and artificial scanpaths can be improved by adding further biological constraints to neural networks. Furthermore, the scanpaths generated by more complex task models need to be analyzed. More complex task models could include object detectors, segmentation models, and visual transformers. Lastly, further work should investigate if NeVA can create highly task-specific scanpaths that resemble those of a human experts (i.e., by using a network trained to detect tumors to generate scanpaths for a medical application). These scanpaths could be used to constrain the attention of neural networks to relevant image regions and thus improve the computational load.

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

# A   Example scanpaths

To provide a better intution about the outcome of the NeVA algorithm, here we provide some illustrations. Figure 2 shows example scanpaths generated by the NeVA algorithm.The first row shows the original stimulus, the second row shows the foveation heatmaps and fixations, and the last row shows the internal representation after the last fixation.

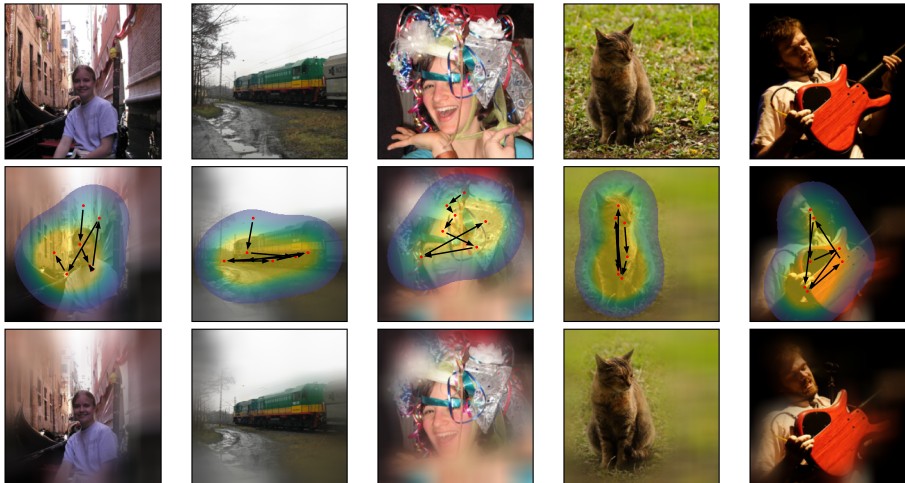

Figure 2: Example scanpaths generated by NeVA.

