# OpenReview forum: "Simulating Human Gaze with Neural Visual Attention"
_NeurIPS.cc/2022/Workshop/GMML — Gaze Meets ML 2022 Oral_

### Official Review · Reviewer_Jtit · 2022-10-09
**Task-directed, top-down modeling for state of the art prediction of scanpath**

**Rating:** 9
**Confidence:** 4

**Review:**

# Quality
The approach is sound. I would recommend that the authors consider models other than ResNets that are trained on CIFAR-10. In particular, the CIFAR-10 dataset only uses 10 classes with images of 32x32 pixel resolution. Datasets such as ImageNet (thousands of classes) and COCO (high resolution, multiple objectives-- classification, segmentation, captioning) may provide additional evidence that the scanpath is indeed influenced by the top-down task.

# Clarity
It is well-written and describes the methods, results, and conclusions in sufficient detail.

# Originality
The authors developed a novel, top-down approach to modeling scanpath as a series of foveations which cumulatively lead to the optimization of the task. Previous models are based on bottom-up approaches that consider the saliency of different regions in the image as the primary influence to the scanpath. The authors posit that task/goal completion is the causal factor instead. Goal-directed movement has been shown in neurophysiological studies beginning with the neural population vector encoding studies of Georgopoulos. The authors use a creative combination of reinforcement learning with pre-trained goal/task-directed neural networks to develop a new class of scanpath models.

> Science 233, 1416 (1986); AP Georgopoulos, et al. Neuronal population coding of movement direction

# Significance
The approach is a new state of the art in scanpath prediction based on its comparison to previous models on 3 standard datasets. It is biologically plausible and leads to immediate questions as to the effect of variations of task demands on scanpath, such as task difficultly, attention span, and working memory. A recent relevant reference is
>Schwetlick, L., Backhaus, D., & Engbert, R. (2022). A dynamical scan-path model for task-dependence during scene viewing. Psychological Review. Advance online publication. https://doi.org/10.1037/rev0000379

# Pros
* Well written, straightforward approach
* Biologically plausible, top down approach to modeling scanpath
* Task-directed goal hypothesis leads to several testable variations in future experiments (task difficultly, subject attention span)

# Cons
* Would like to see the results from different task models (e.g. visual transformers) and additional task classes.

---

### Official Review · Reviewer_nx1v · 2022-10-12
**Simulation of scanning patterns by using Neural Visual Attention algorithm.**

**Rating:** 7
**Confidence:** 3

**Review:**

The paper studies a biologically plausible way of simulating human scanning patterns with the Neural Visual Attention algorithm. The authors evaluate their work with three different datasets (i.e., MIT1003, TORONTO, and KOOSTSTRA) and their approaches mostly outperform the previous work and work better than the random baselines for free viewing task.

## Quality
The technical approach seems to be valid. Evaluation of the approach also shows that their methods outperform previous works and random baseline. However, looking at the SPP results, there is a lot of room for improvement to the human baseline. In addition, I would be very interested in seeing how methods perform on specific tasks other than free viewing since, in the expertise domain, users usually perform a specific task for that domain.

## Clarity
The paper is well-organized and nicely written.

## Originality
The authors' proposal using a task model, foveation layer, and an attention mechanism seems to be novel.

## Significance
Evaluations by using three different datasets and outperforming previous works in these show a significant push for simulating the scanpaths. However, I do not agree with the authors' argument that NeVA can be used for low-cost synthetic data generation in the discussion, since they also stated that the human baseline outperforms all the others by a large margin.

---

### Official Review · Reviewer_1fJc · 2022-10-16
**Clearly described model and experiments.**

**Rating:** 8
**Confidence:** 3

**Review:**

The paper describes an approach to generate realistic task-specific Gaze behaviour built on top of DL-based  classification  and reconstruction tasks.
The model they propose uses a Gaussian to decreases high-frequency information in an image away from the focus point. The results demonstrate that the classification task provides better ground for generating focus similar to human visual behaviour. The results are interesting and they provide strong support for their work. I am not sure if the comparison with third-part models is exhaustive and I would like the authors or other reviewers to confirm that.
It would also be interesting to see what the effect of the attention mechanism are to the tasks. I suspect the effect is negative and therefore the authors avoid commenting.
I believe there is an error in the results claiming that G-Eymol gives the best result in one task and CLE the second best. Table 1 suggests that CLE gives the best result in SPP SBTDE so either the Table or the next has an error.

---

### Meta-Review · Area_Chair_L69G · 2022-10-20

**Recommendation:** Accept (Oral)
**Confidence:** 5

**Metareview:**

This work proposes a Neural Visual Attention (NeVA) algorithm that simulates biologically plausible human-like human scanning patterns. The paper is well-organized and provides sufficient experimental results on three datasets. The reviewers seem to agree on the novelty and the strong support of the paper's contributions as indicated by the competitive results w.r.t. baselines. Reviewers have provided positive feedback with further revision suggestions to improve the quality of the work, such as qualitative analysis of the attention, improving the human baseline, and experiments with larger datasets. Based on the review ratings and comments, I recommend accepting and believe this work would be of general interest to the workshop audience.

---

### Decision · Program_Chairs · 2022-10-20

Accept (Oral)